# Myosin Motors: Novel Regulators and Therapeutic Targets in Colorectal Cancer

**DOI:** 10.3390/cancers13040741

**Published:** 2021-02-11

**Authors:** Nayden G. Naydenov, Susana Lechuga, Emina H. Huang, Andrei I. Ivanov

**Affiliations:** 1Department of Inflammation and Immunity, Lerner Research Institute, Cleveland Clinic Foundation, Cleveland, OH 44195, USA; naydenn@ccf.org (N.G.N.); lechugs@ccf.org (S.L.); 2Departments of Cancer Biology and Colorectal Surgery, Cleveland Clinic Foundation, Cleveland, OH 44195, USA; huange2@ccf.org

**Keywords:** actin cytoskeleton, motor proteins, tumor growth, vesicle trafficking, matrix adhesion, migration, invasion, metastasis

## Abstract

**Simple Summary:**

Colorectal cancer (CRC) is a deadly disease that may go undiagnosed until it presents at an advanced metastatic stage for which few interventions are available. The development and metastatic spread of CRC is driven by remodeling of the actin cytoskeleton in cancer cells. Myosins represent a large family of actin motor proteins that play key roles in regulating actin cytoskeleton architecture and dynamics. Different myosins can move and cross-link actin filaments, attach them to the membrane organelles and translocate vesicles along the actin filaments. These diverse activities determine the key roles of myosins in regulating cell proliferation, differentiation and motility. Either mutations or the altered expression of different myosins have been well-documented in CRC; however, the roles of these actin motors in colon cancer development remain poorly understood. The present review aims at summarizing the evidence that implicate myosin motors in regulating CRC growth and metastasis and discusses the mechanisms underlying the oncogenic and tumor-suppressing activities of myosins.

**Abstract:**

Colorectal cancer (CRC) remains the third most common cause of cancer and the second most common cause of cancer deaths worldwide. Clinicians are largely faced with advanced and metastatic disease for which few interventions are available. One poorly understood aspect of CRC involves altered organization of the actin cytoskeleton, especially at the metastatic stage of the disease. Myosin motors are crucial regulators of actin cytoskeletal architecture and remodeling. They act as mechanosensors of the tumor environments and control key cellular processes linked to oncogenesis, including cell division, extracellular matrix adhesion and tissue invasion. Different myosins play either oncogenic or tumor suppressor roles in breast, lung and prostate cancer; however, little is known about their functions in CRC. This review focuses on the functional roles of myosins in colon cancer development. We discuss the most studied class of myosins, class II (conventional) myosins, as well as several classes (I, V, VI, X and XVIII) of unconventional myosins that have been linked to CRC development. Altered expression and mutations of these motors in clinical tumor samples and their roles in CRC growth and metastasis are described. We also evaluate the potential of using small molecular modulators of myosin activity to develop novel anticancer therapies.

## 1. Introduction

Colorectal cancer (CRC) is the most commonly diagnosed gastrointestinal cancer, with a high mortality rate and rising incidence in young adults [1,2,3]. The etiology of CRC involves a complex crosstalk between genetic and environmental factors. The genotypic and phenotypic progression from normal mucosa to polyp to invasive cancer, known as the “adenoma to carcinoma sequence”, was pointed out decades ago [4]. The genetic abnormalities in CRC are characterized by high-frequency mutations in key oncogenic and tumor suppressing genes, such as Kirsten Rat Sarcoma Viral Proto-Oncogene (KRAS), p53 and Adenomatous Polyposis Coli (APC) [5,6]. The environmental factors contributing to CRC development include diet, obesity, infection, tobacco use and alcohol consumption [7]. Sporadic CRC originates in differentiated intestinal epithelial cells, which then undergo a remarkable phenotypic plasticity during tumor progression, resulting in acquisition of a highly invasive, metastatic cell phenotype [8,9]. While rates of cure for the early stage disease may be above 90%, for the locally advanced and metastatic disease, 5-year survival is dismal, and may be as low as 15% [3]. Importantly, mechanisms of the advanced, widely spread metastatic CRC remain poorly understood. For this reason, effective anti-metastatic therapy has yet to be developed.

Remodeling of the actin cytoskeleton is a key driver of phenotypic plasticity and metastasis in different cancers [10,11,12]. This cytoskeleton represents an elaborate network of actin filaments with intrinsic polarity and dynamics. Actin filaments are constantly turning over by adding actin monomers to their growing plus end and removing the monomers from the opposite minus end. Additionally, the actin filaments readily translocate to and self-associate and interact with intracellular organelles, thereby creating a diverse and dynamic cellular environment. Remodeling of actin filaments controls two major cellular processes involved in tumor development; specifically, cell division that fuels tumor growth and cell migration that drives metastasis. The functional consequences of oncogenic mutations of *KRAS*, p53 and *APC* in CRC involve marked alterations of the actin cytoskeleton [13,14,15]. Furthermore, the abnormal structure and dynamics of actin filaments could be induced by altered physical properties and increased proinflammatory signaling of the tumor stroma [12,16,17,18]. Such a complex interplay between autonomous and extrinsic factors in the cell creates significant challenges for understanding the mechanisms and functional consequences of the altered actin cytoskeletal organization in CRC.

Actin filaments have limited ability to self-remodel, and their assembly and turnover in living cells is aided by a large number of actin-binding proteins with specific molecular properties and functions [19,20]. Among them, myosins play prominent roles in regulating cytoskeletal organization and dynamics. Myosins form a large superfamily of evolutionary conserved actin-binding motor proteins that converts chemical energy of the ATP hydrolysis to generate force and movement along actin filaments [21,22,23]. There are 40 genes encoding different myosins in the human genome, divided into 13 different classes based on their sequence and structural characteristics [24,25]. Class II was the first discovered and the most characterized class of myosins; it mediates muscle contraction and regulates contractile processes in non-muscle cells [23,24]. Members of this class are called “conventional” myosins, whereas myosins belonging to all other classes are referred to as “unconventional” myosins [23,26]. All known members of the myosin superfamily are built following the same structural “blueprint” [21,22,27]. They have a conserved N-terminal motor domain that interacts with actin and possesses actin-activated ATPase activity (Figure 1). The motor domain is responsible for myosin movement along the actin filament driven by the chemical energy released from ATP hydrolysis. The actual movement of myosin molecules originates from the cycle of the motor domain binding to and dissociating from the actin filaments, and depends on interactions with ATP, ADP and phosphate generated during ATP hydrolysis [21,23]. Furthermore, such movement is driven by small conformational changes in the motor domain generated by phosphate release [21,23]. The myosin motor domain is followed by a helical extension that contains several binding sites for calmodulin or calmodulin-like myosin light chains and is called a “lever arm”. The lever arm amplifies movement caused by conformational changes in the motor domain [21]. Finally, myosins possess a C-terminal tail of variable length, characterized by several domains and structural motifs responsible for interactions with different proteins and lipid membranes [23,26]. The C-terminal tail is the least conserved part of the myosin molecule, determining the unique binding properties and functional features of the different myosin classes. While conventional myosins act as classic actin motors that slide actin filaments against each other (Figure 1A), unconventional myosins mediate the transport of proteins and membrane vesicles along actin filaments, tether actin filaments to lipid membranes and serve as scaffolds for large multiprotein complexes in the nucleus and cytoplasm [23,26] (Figure 1B-D). A combination of motor and scaffolding abilities determines the key cellular functions of myosins in regulating cell division, cell–cell and cell–matrix adhesions, cell migration, vesicle trafficking and cell differentiation [21,23,26,28,29]. Myosin motors are increasingly recognized as essential regulators of tumorigenesis, playing either oncogenic or tumor-suppressing roles [30,31,32,33]. Because of the complexity of myosin functions and upstream regulatory mechanisms, their roles in tumorigenesis remain controversial and poorly understood. Furthermore, the majority of studies focusing on myosins have been conducted in breast, lung and prostate cancers [30,31,32,33]. However, relatively little is known about how these cytoskeletal motors regulate CRC development and metastasis. The present review addresses this important subject. We will summarize the available data about mutations and altered expression of different members of the myosin superfamily in CRC tissues and will discuss how altered activity of myosins could regulate colon cancer cell growth and migration. We will also discuss the potential of using small molecular modulators of myosin activity to develop novel anticancer therapies.

## 2. Conventional Myosins

### 2.1. Activation Status of Conventional Myosins in Colon Cancer

Class II, also known as conventional myosins, is the biggest subfamily, and includes skeletal, cardiac, smooth muscle and non-muscle myosins [34]. They function as hexamers consisting of two heavy chains, two essential light chains and two regulatory myosin light chains (RMLC) [23,28]. Light chains bind to specific sites at the lever arm and regulate myosin heavy chain conformation and interactions. Phosphorylation of RMLC by either myosin light chain kinase (MLCK) or Rho-associated kinase (ROCK) is considered an obligatory mechanism driving activation of conventional myosins [28,35,36]. A characteristic feature of all conventional myosins is their ability to form bipolar myofilaments via self-assembly of their C-terminal α-helical coiled-coil tails (Figure 1A) [37,38,39]. Interactions of such myofilaments with actin filaments results in generating contractile forces and actin filament cross-linking into thick actomyosin bundles [23,38,39]. In order to understand the functional roles of conventional myosins in colon cancer, it is critical to determine the activation status of these cytoskeletal motors. This is a complicated task given the multiplicity of mechanisms that can regulate their activity. The most important mechanisms include expression of myosin heavy chains, phosphorylation of RMLC and assembly of high order myofilaments, which could be modulated by either myosin heavy chains phosphorylation or their interactions with different accessory proteins [28,37,38,39]. While systematic studies of the activity of conventional myosins in colon cancer have not been undertaken, published evidence, discussed below, nevertheless suggest that these motors could be either activated or inhibited in CRC.

### 2.2. Mechanisms Underlying Activation of Conventional Myosins in Colon Cancer

*Upregulation of non-muscle myosin IIA expression*: Non-muscle myosin II (NM II) is an extensively studied cytoskeletal motor that is ubiquitously expressed in non-muscle tissues, including intestinal epithelium [28,38,39,40]. Given the epithelial origin of CRC, NM II is likely to be the most important conventional myosin involved in colonic tumorigenesis. CRC cells express three NM II paralogs, NM IIA, NM IIB and NM IIC, encoded by different genes (*MYH9*, *MYH10* and *MYH14*, respectively) [41,42]; however, the literature to date is limited to examining NM IIA expression in colon cancer (Table 1).

Thus, NM IIA mRNA expression was found to be significantly upregulated in CRC tissue [43]. Interestingly, in sporadic CRC, characterized by liver metastasis (stage IV), a high NM IIA mRNA level was associated with an A769T point mutation in the myosin motor domain [44]. Agreeing with the mRNA expression data, a recent study of a small number of patient samples reported the increased level of NM IIA protein in CRC relative to paired non-neoplastic colonic tissues [45]. Especially noteworthy is that Kaplan–Meier curves demonstrate that high NM IIA expression is significantly associated with poor overall survival and disease-free survival of CRC patients [44,45], thereby highlighting this motor as a potential contributor to colonic carcinogenesis. While the exact localization of upregulated NM IIA in colonic tumor tissue remains to be investigated, evidence to date suggests that this motor could be overexpressed in both cancer cell and stromal compartments. Thus, many human colon cancer cell lines express an elevated level of the NM IIA protein [41,42,45]. Furthermore, a laser capture microdissection study observed upregulation of the NM IIA protein expression in stromal cells isolated from human colon adenocarcinoma [46]. Increased NM II expression in cancer-associated stromal cells could represent an important oncogenic mechanism, given the reported data from a mouse model of breast cancer progression, in which increased expression of NM IIA, NM IIB and RMLC was observed in cancer-associated fibroblasts (CAF) during transition from the hyperplasia to the adenoma tumor stages [61]. Such NM II upregulation and increased actomyosin contractility were essential for stiffening of the extracellular matrix and CAF activation that was driven by a mechanosensitive transcriptional factor, YAP [61]. While molecular mechanisms underlying the increased expression of NM II in CRC remain to be investigated, the inflammatory environment of the tumor stroma, which is particularly important in colitis-associated colon cancer (CAC), could be responsible for the elevated level of this cytoskeletal motor. This suggestion is supported by a recent proteomic study that found marked upregulation of NM IIA expression in colonic epithelial cells isolated from resected tissues of ulcerative colitis patients [62].

*Activating mutations and overexpression of MYH11*: High-frequency frameshift mutations in another conventional myosin gene, *MYH11*, have been described in CRC characterized by microsatellite instability (Table 1) [47,49,63]. *MYH11* encodes two splice variants of a smooth muscle myosin heavy chain, SM1 and SM2, and the frameshift CRC-associated mutations occur in the C-terminal tail domain of the SM2 isoform [47]. These mutations have an unexpected functional effect, in that they eliminate the RMLC phosphorylation-dependent regulation of myosin ATPase activity, thereby creating a constitutively active SM2 motor [47]. Interestingly, an ATPase-activating *myh11* mutation was shown to cause the *meltdown* (*mlt*) phenotype in zebrafish [64,65]. This phenotype is characterized by uncontrolled activation of smooth muscles surrounding the zebrafish intestine that triggers proliferation and invasion of intestinal epithelial cells [64,65]. It is unclear how such stromal-to-epithelial signaling in the *mlt* zebrafish mutant is related to the effects of activating *MYH11* mutations in CRC patients, since human mutations were localized in the epithelial, not the stromal tumor compartment [47]. In addition to acquiring activating mutations, SM2 expression was found to be upregulated in CRC [48,50], and such upregulation correlated with poor patient survival [48].

*Phosphorylation of the regulatory myosin light chains*: Since RMLC phosphorylation is a critical mechanism of NM II activation, it is commonly used as a biochemical readout of myosin motor activity. It is surprising, therefore, that we cannot find published studies examining the level of RMLC phosphorylation in human CRC tissues. The lack of published studies is likely due to technical difficulties in preserving and detecting the phospho-RMLC signal in clinical samples. However, in two mouse models, such as spontaneous CRC in APC^Min/+^ mice and azoxymethane–dextran sodium sulfate (AOM-DSS)-induced CAC, the phospho-RMLC level was markedly elevated in tumor tissue as compared to normal intestinal mucosa [66,67]. In addition, evidence of activation of the RhoA/ROCK signaling axis in CRC suggests that RMLC phosphorylation could be increased during colonic tumorigenesis [17,68]. RhoA/ROCK signaling is a multi-modal regulator of the actin cytoskeleton, with RMLC being a major downstream effector of this signaling cascade [35,36]. Several mechanisms contribute to RhoA/ROCK activation in CRC. They include upregulation of RhoA and ROCK1 expression [69,70,71], acquiring activating mutations of ROCK1 in microsatellite-instable colon cancer [72], as well as frequent inactivating mutations of upstream negative regulators of RhoA, Rho GTPase-activating proteins [73]. Interestingly, expression of the *MYLK* gene that encodes MLCK was shown to be downregulated in CRC [74,75]. This data suggests a possible switch from MLCK to ROCK-dependent RMLC phosphorylation in colon cancer cells. While MLCK and ROCK target the same Ser/Thr residues in RMLC, they are known to phosphorylate and activate topographically distinct actomyosin structures in epithelial cells and fibroblasts [76,77,78]. It is possible that switching from the MLCK to the Rho/ROCK-dependent modes of regulation of NM II phosphorylation and activity could have significant effects on the growth and invasion ability of colon cancer cells.

### 2.3. Mechanisms of Conventional Myosins Inactivation in Colon Cancer

*Activating K-RAS mutations*: Colon cancer is characterized by high-frequency mutations of the *K-RAS* oncogene [5,6]. The majority of these mutations (such as a G13D mutation) result in a constitutively active K-Ras protein. Studies in colon cancer cell lines demonstrated that the active K-RAS G13D mutant causes a profound disruption of actomyosin structures, such as stress fibers, and decreases NM II activity even in the presence of a high level of active RhoA [15,79,80]. While the underlying mechanisms remain poorly understood, at least two possibilities have been suggested. One is that K-Ras dependent activation of mitogen-activated kinase (MAP) signaling uncouples RhoA activity from its downstream cytoskeletal targets [15]. The other mechanism involves stimulation of Rac1 GTPase that antagonizes RhoA activity, thereby decreasing RMLC phosphorylation and causing NM II inactivation [80].

*Overexpression of S100 proteins*: A different mechanism that targets NM II heavy chains could contribute to functional inactivation of NM II in CRC. This mechanism involves overexpression of pro-metastatic proteins of the S100 family. The S100 family members are multifunctional calcium-binding proteins with important roles in cancer growth and metastasis [81,82,83]. While these proteins exhibit multiple cellular activities, one of their important functions involves interactions with NM II heavy chains and regulation of myofilament assembly. Multiple S100 isoforms were shown to interact with NM II [84]; of these, S100A4 and S100P interactions are more extensively characterized. S100A4 and S100P bind to the C-terminal tailpiece of the NM II heavy chain, having higher affinity to NM IIA and NM IIC compared to the NM IIB isoforms [84,85,86,87]. Furthermore, experiments in cell-free systems demonstrate that these S100 isoforms efficiently disassemble NM IIA and NM IIC myofilaments, but have little effect on NM IIB filament assembly [84,85,86,87]. Importantly, pro-motile effects of S100A4 in cancer cells depend on its interactions with NM IIA [88,89]. S100A4 and S100P are upregulated in CRC and their high expression correlates with poor patient survival [82,90,91,92]. Hence, it would be important to elucidate the interplay between upregulated S100 proteins and NM II functions in colon cancer.

### 2.4. Functional Roles of Conventional Myosins in Colon Cancer

*Regulation of cancer cell proliferation*: Existing evidence implicate certain conventional myosin motors in regulating two key oncogenic features of CRC, such as tumor growth and metastatic dissemination (Figure 2).

For example, the roles of NM IIA in controlling colon cancer growth has been recently demonstrated using knockdown and overexpression of this motor in canonical CRC cell lines, such as SW620, SW480 and HCT116 cells [45,93]. These studies indicate that NM IIA depletion attenuates both anchorage-dependent and soft agar growth of CRC cells in vitro, whereas its overexpression has growth-promoting effects. Furthermore, high NM IIA expression significantly accelerates growth of CRC xenografts injected into nude mice [45]. The described tumor growth-promoting effects of NM IIA are consistent with studies in other tumor types; specifically breast, pancreatic cancer and hepatocellular carcinoma, in which lowering NM IIA expression attenuates growth of cancer xenografts in mice [94,95,96]. Moreover, loss of the NM II activator, RhoA, also inhibits the growth of CRC cells engrafted in nude mice [97]. In contrast to the tumor growth-promoting effects of NM IIA observed in mouse tumor xenograft models, its opposite, anti-proliferative and tumor suppressive functions have been reported in transgenic mouse models of spontaneous tumorigenesis. For example, either NM IIA knockdown by RNA interference or NM IIA knockout in mice results in the development of squamous cell carcinoma on tumor-susceptible backgrounds [98]. Likewise, deletion of NM IIA triggers formation of squamous cell carcinoma in tongue epithelium [99]. Finally, a mosaic loss of either NM IIA or NM IIB causes excessive epithelial proliferation in mouse mammary glands [100]. While the reasons for such conflicting functional effects of NM IIA on tumor growth in mouse xenograft and spontaneous oncogenesis models remain unknown, they could lie in fundamental differences in the biophysical environment for tumor development in these models. Thus, a recent study of oncogene-driven induction of pancreatic neoplasia discovers that tissue geometry, such as the diameter, curvature and deformation of epithelial tubes, profoundly affects the development of neoplastic lesions [101]. Such effects of the physical environment on tumor outgrowth depend on changes in the distribution of active NM II [101]. This data suggests that NM II acts as an early sensor of the physical environment surrounding tumor-initiating cells and a mechanotransducer that instructs tumor development. Differences in the biophysical properties of the surrounding tissues for tumor xenografts and epithelial tumors that spontaneously develop in “natural” tissue environments are likely to be responsible for the reported contrasting effects of NM II activity on tumor development. Another reason that is especially relevant to colon cancer is the NM II-dependent effects on the intestinal epithelial barrier and tissue inflammation. Indeed, CRC development is driven by an inflammatory milieu, which is established through activation of the intestinal mucosal immune system by gut microbiota [7]. The extent of this inflammation is regulated by the epithelial barrier that limits host–pathogen interaction. NM II is a known regulator of the gut barrier [40,102] and both activation and inactivation of NM II result in barrier disruption and mucosal inflammation in vivo [103,104]. Hence, altered myosin activity could affect tumor development by controlling the epithelial barrier integrity and tissue inflammation in spontaneous models of CRC, but not under conditions of sterile xenograft growth. It is important to note that no studies have investigated the effects of NM II knockout or overexpression in CRC development using pathophysiology-relevant models with CRC susceptible backgrounds.

Little is known about the mechanisms by which NM II controls colon cancer growth. One possibility lies in the regulation of colon cancer stem cells. Indeed, pharmacologic inhibition of either NM II or ROCK was shown to promote expansion of colon cancer initiating cells by upregulating expression of a stem cell marker, CD44 [105]. Likewise, a monoallelic deletion of NM IIA in mouse intestinal epithelium improves survival and growth of intestinal stem cells [106]. This data is in line with several reports highlighting the role of NM II in modulating differentiation of embryonic and induced pluripotent stem cells in different tissues [107,108,109]. Such NM II-dependent regulation of cell stemness and differentiation is likely to be linked to its ability to sense mechanical forces and modulate assembly of the cytoskeletal structures and signaling that eventuates in the altered nuclear transcriptional program [94,95,108].

*Regulation of cancer cell invasion and metastasis*: Since metastatic dissemination of CRC is the primary reason for high patient mortality, it is essential to understand the mechanisms underlying invasion and metastasis of colon cancer cells [110]. Modulation of CRC cell migration and invasion in vitro by using pharmacologic and genetic NM II inhibition reveals either pro- or anti-migratory roles of this actin motor [41,45,111]. For example, pharmacologic inhibition of NM II with blebbistatin, as well as siRNA-mediated downregulation of NM IIA expression inhibits wound healing in well-differentiated SK-CO15 colonic adenocarcinoma cell monolayers [41]. This data is consistent with a current paradigm considering NM II as a key organizer of the planar cell motility [39,112,113]. By contrast, pharmacologic or genetic inhibition of NM IIA markedly stimulates SK-CO15 cell invasion into Matrigel [41]. Likewise, Caco-2 colon cancer cells embedded into 3-D Matrigel form well-polarized spherical cysts with a smooth periphery lacking robust cell protrusions [111,114]. Inhibition of either NM II or its upstream activator, ROCK, induces outgrowth of actin-rich protrusions from the Caco-2 cyst surface, eventuating in the collective cancer cell invasion of the surrounding matrix [111,114]. Consistent with these findings, a pharmacologic activator of actomyosin-dependent contractility, 4-hydroxyacetophenone, specifically targeting the NM IIB and NM IIC isoforms, inhibits invasion of HCT116 cells in vitro and attenuates metastatic spread of HCT116 cell xenografts in nude mice [115]. The described conflicting roles of NM II in regulating planar migration and 3-D hydrogel invasion of CRC cells likely reflect distinct functions of this cytoskeletal motor under different physical properties of the cell environment. Thus, a classic wound healing assay investigates cell migration over a very stiff substratum, such as glass or plastic. This process depends on the formation and remodeling of specialized matrix adhesion structures, focal adhesions (FA), driven by high tensile forces generated by FA-associated basal actomyosin stress fibers [113,116,117]. NM II inhibition disrupts stress fiber and FA assembly, thereby attenuating cell interactions with extracellular matrix and directional cell motility [41]. In contrast, cancer cell invasion into soft 3-D gels does not involve assembly of robust FA and may require rapid remodeling of the cortical actomyosin cytoskeleton to accelerate cell squeezing through the porous matrix [118,119]. In this scenario, NM II inhibition may promote cell deformability, thereby accelerating cell migration through the confined spaces. Furthermore, decreased physical forces acting upon cancer cells during Matrigel invasion could unmask NM II-dependent signaling mechanisms that are essential for cell invasion. One such mechanism involves stimulation of the MAP kinase/ERK signaling axis that was shown to promote invasion of NM II-inhibited CRC and glioma cells [41,120]. Since CRC development is accompanied by alterations in physical properties of the tissue environment toward a stiffer stroma, it is unclear how altered NM II activity in colon cancer cells would modulate their matrix invasion and metastasis in vivo. Future studies on the manipulation of NM II expression in intestinal epithelium and stroma in mice having genetic backgrounds, which are susceptible to colon cancer, are needed to address this important question.

## 3. Unconventional Myosins

### 3.1. Class I Myosins

Unconventional class I myosins is a family of widely expressed cytoskeletal proteins that in the human genome is encoded by eight different genes (*MYO1A* through *MYO1H*). These are monomeric motors that do not form filaments but have the ability to interact with membrane phospholipids via a pleckstrin homology domain in their C-terminal tail [23,121]. Due to a dual affinity to actin and membrane phospholipids, class I myosins tether actin filaments to the cellular membranes [23,121]. These molecular features determine the broad functions of class I myosins in regulating membrane tension, cell–cell and cell–matrix adhesions and intracellular vesicle trafficking [23,121]. Two members of this protein family, unconventional myosin-1A (MYO1A) and myosin-1D (MYO1D), have been implicated in CRC (Table 1, Figure 2). MYO1A is an essential cytoskeletal component of differentiated enterocytes that is especially enriched in brush border microvilli [122]. Frequent frameshift mutations of MYO1A were observed in CRC tumors with microsatellite instability [51]. Some of these mutations alter the amino acid sequence on the tail domain of MYO1A, impairing its interactions with plasma membrane and causing mislocalization and decreased stability of this motor [51,123]. Another mechanism leading to the impairment of MYO1A functions in colon cancer cells involves promoter hypermethylation, which downregulates expression of this motor [123]. Importantly, low MYO1A expression in CRC was shown to be a prognostic marker of poor patient survival [51]. Either depletion of MYO1A or overexpression of its CRC-relevant mutant result in loss of the apico-basal cell polarity and dedifferentiation of colonic epithelial cells in vitro (Figure 2). Such dedifferentiation accelerates the soft agar growth of colon cancer cells [51]. While loss of MYO1A expression in mice appears to be insufficient to induce spontaneous tumorigenesis, MYO1A null animals display higher susceptibility to induced intestinal tumorigenesis, both in genetic (APC^Min/+^) and chemical (AOM-DSS) models of murine CRC [51]. Despite the reported association of MYO1A downregulation and tumor cell invasion of local tissues in human CRC samples, deletion of this motor does not promote a metastatic phenotype in the APC^Min/+^ CRC model [51]. Future studies are needed to determine whether inactivation of MYO1A plays an important role in metastatic dissemination of CRC tumors.

A recent study implicated another member of the myosin I family, MYO1D, in the regulation of CRC development [52]. In contrast to MYO1A, MYO1D protein expression was found to be upregulated in the advanced stages III and IV of CRC, although its association with patient survival has not been established [52]. Overexpression of MYO1D accelerates CRC cell invasion in vitro and promotes tumorigenesis in a syngeneic mouse xenograft model [52]. Mechanisms of tumor-promoting activities of MYO1D are linked to regulation of growth factor receptor trafficking. Specifically, MYO1D was shown to directly interact with epidermal growth factor receptor (EGFR) and promote retention of this receptor at the plasma membrane. Such prolonged plasma membrane accumulation accelerated EGFR activity in MYO1D-overexpressing CRC cells, thereby increasing their tumorigenic features [52]. This data suggests that either lowering overexpression or inhibiting activity of MYO1D could have therapeutic potential in a subtype of CRC with high expression of EGFR family members and could also help to overcome tumor resistance to anti-EGFR therapy.

### 3.2. Class V Myosins

Myosin V is an ancient subfamily of unconventional myosins that form two-headed motors and possess a distal globular tail domain involved in interactions with different adaptor proteins [23,26,29]. The majority of known myosin V adaptors are members of the Rab family small GTPases playing key roles in intracellular vesicle trafficking [124]. Interactions with different Rab proteins determine functions of myosin V motors in regulating long-distance trafficking of protein, RNA and lipid cargo, as well as organelle translocation [23,26,29]. This myosin class has three members, myosin-5A (MYO5A), myosin-5B (MYO5B) and myosin-5C (MYO5C), with a high degree (60–80%) of sequence similarity; these members may play either unique or redundant cellular roles [23,26,29]. Among the class V myosin motors, MYO5A and MYO5B have been implicated in CRC development (Table 1, Figure 2). The first study linking these unconventional myosins to colon cancer reported increased MYO5A expression in colon cancer, characterized by lymph nodes or distant metastasis (CRC stages III and IV) in contrast to non-metastatic CRC [53]. Moreover, MYO5A expression was found upregulated in highly invasive CRC cell lines, Lovo and SW480, where MYO5A overexpression was driven by a pro-metastatic transcriptional factor, Snail [53]. Functional studies demonstrated that MYO5A depletion impairs migration of different colon cancer cells in vitro and attenuates metastatic spread of these cells in a chicken chorioallantoic membrane assay in vivo [53]. While no subsequent studies addressed the mechanisms of MYO5A-driven CRC progression, depletion of this motor in HeLa cells and melanoma cells is also inhibits cell migration and invasion [125,126]. Cumulatively, this data strongly suggests that MYO5A acts as an important driver of metastatic dissemination of various tumors.

Unlike MYO5A, its close homolog, MYO5B, appears to act as a tumor suppressor. MYO5B was shown to be downregulated at the mRNA and protein levels in primary CRC samples [54]. This study observed a gradual reduction in MYO5B protein expression depending on the tumor grade, thus being higher in well-differentiated and lower in poorly differentiated tumors (histologic grades I and III, respectively). Furthermore, MYO5B was signified as a prognostic marker for CRC with its lower expression being associated with poor overall and disease-free survival of cancer patients [54]. Despite such promising clinical data, functional consequences of MYO5B downregulation in CRC have not been investigated. In normal intestinal epithelium either loss-of-function mutations or deletion of this motor induces a microvillar inclusion disease phenotype characterized by lack of apical microvilli and accumulation of intracellular vacuoles containing apical plasma membrane structures [127,128]. These abnormalities reflect disruption of global vesicle trafficking, leading to a loss of the apico-basal cell polarity in epithelial cells with defective MYO5B functions. It is noteworthy that decreased MYO5B expression was observed in gastric cancer [129], whereas missense mutations of this motor protein were associated with rare neural tumors, pheochromocytoma and paraganglioma [130]. Functional studies revealed that MYO5B depletion markedly stimulates migration and Matrigel invasion of gastric cancer cells [129]. Furthermore, overexpression of MYO5B mutants, characteristic of neural tumors, accelerates cancer cell migration along with stimulation of cell proliferation [130]. The reportedly contrasting functional roles of MYO5A and MYO5B in cancer are puzzling, given the high sequence and structural similarity of these motors. In fact, the high conservation between cargo-binding distal globular tail domains of mammalian class V myosins suggests that they participate in similar trafficking events [131]. Redundant functions of these motors in regulating neuronal cell growth have been reported [132]. Interestingly, both MYO5A and MYO5B were found to interact with a key tumor suppressor, phosphatase and tensin homolog (PTEN), in neural cells [132]; however, whether PTEN trafficking underlies functions of either MYO5A or MYO5B motors in cancer cells remains unknown.

### 3.3. Class VI Myosins

Myosin-6 (MYO6), the only mammalian member of the unconventional class VI myosins, is a unique cytoskeletal motor that moves cargo toward the minus end of the actin filament [21]. This feature allows MYO6 to move molecules or vesicles from the plasma membrane toward the cell interior, thereby playing essential roles in regulating endocytosis [23,26,29]. This cytoskeletal motor also controls additional intracellular trafficking events, such as exocytosis and autophagy [26,29,133]. The complexity of MYO6 functions is further illustrated by its localization in the nucleus, where it participates in the regulation of gene expression [134,135].

Several recent studies report upregulation of MYO6 expression in CRC samples (Table 1) [55,56,57,134,135]. This is consistent with the increased MYO6 level observed in ovarian carcinoma and prostate cancers, and signifies that this cytoskeletal motor is a potential tumor promoter [31,32]. Importantly, a high MYO6 level is prognostic of poor survival of CRC patients [56]. Increased MYO6 expression in colon cancer is regulated by a complex interplay between microRNAs (miR) and long non-coding RNAs (lncRNA). Specifically, upregulation of SOX21-AS1, urothelial carcinoma associated 1 and hsa_circ_0000231 lncRNAs was found to correlate with high MYO6 expression in CRC tissues [55,56,136]. Studies utilizing CRC cell lines demonstrated that these lncRNAs stimulate MYO6 expression by acting as sponges for MYO6-suppressing miR-502, miR-145 and miR-143 [55,56,136]. Little is known about the functional consequences of MYO6 overexpression in colon cancer cells, where it appears to have pro-growth and pro-survival activities (Figure 2) [57]. The pro-survival activity of this motor is conserved among various cancer types and such MYO6 activity could be linked to stabilization and activation of p53 [137,138]. Furthermore, cell-growth promotion by MYO6 is associated with its known interactions with RNA polymerase II in the nucleus and stimulation of mRNA transcription [134,135], although the causal connection of this mechanism to cancer growth has not been investigated. In addition to stimulating cancer cell proliferation, MYO6 promotes migration and invasion of various cancer cells by mechanisms that remain to be identified [139,140,141].

### 3.4. Class X Myosins

Myosin-10 (MYO10) is a vertebrate-specific motor possessing several scaffolding domains at its C-terminal tail. Among them, pleckstrin homology domains mediate MYO10 interactions with membrane phospholipids, whereas a myosin tail homology domain and a FERM domain regulate its binding with various protein partners [142,143]. MYO10 is usually enriched in the finger-like cellular protrusions, filopodia, and is essential for their formation via diverse mechanisms. Thus, dimerization of MYO10 drives convergence of actin filaments into parallel bundles, providing a core structure for filopodia stability [144]. Furthermore, this motor binds and transports both actin polymerizing proteins, such as Mena and vasodilator-stimulated phosphoprotein, and integrins to the filopidial tips, resulting in their elongation and anchoring into the extracellular matrix [145,146].

MYO10 expression was found elevated in various tumors, most notably in breast cancer and melanoma [143,144,147,148]. In breast carcinoma samples, MYO10 predominantly accumulates at the invasive tumor edges [147]. Consistently, MYO10 depletion inhibits migration and invasion of breast cancer and melanoma cells in vitro and attenuates colonization of the lungs by tumor cells in vivo [144,147,148].

Only one recent study addressed the roles of MYO10 in CRC [58]. This study observed increased MYO10 mRNA expression in colorectal tumors characterized by lymphatic metastasis, in contrast to non-metastatic CRC. In colon cancer cell lines, MYO10 expression was upregulated by signaling through the protease-activated receptor-2, and activity of this motor mediated protease-activated receptor-2 agonist-driven migration of cancer cells [58]. An intriguing but unexplored mechanism of MYO10-dependent regulation of CRC development could be suggested based on observed interactions between this actin motor and important tumor suppressors. In neurons, MYO10 binds to a cell surface receptor called “deleted in colorectal cancer” (DCC) [149]. Furthermore, mutual relationships between DCC and MYO10 have been described, with MYO10 regulating DCC localization at the cell surface and DCC promoting MYO10-dependent filopodia elongation [149,150]. Another MYO10-interacting tumor suppressor is “mutated in colorectal cancer” (MCC), which is a robust modulator of NF-kB signaling in colon cancer cell lines [151]. However, the functional interplay between MYO10 and MCC has not been explored. Since expression of both DCC and MCC is known to be decreased in CRC, it is tempting to speculate that depletion of these binding partners and regulators could markedly affect MYO10 functions in colon cancer cells. This mechanism would be instructive to explore in future studies.

### 3.5. Class XVIII Myosins

Class XVIII myosins is a small subfamily comprising two members, myosin-18A (MYO18A) and myosin-18B (MYO18B). The former protein is abundantly expressed in all studied tissues, whereas the latter paralog is predominantly expressed in cardiac and skeletal muscles [152]. Class XVIII myosins differ from other myosin classes by having large C-terminal and N-terminal sequence extensions [152]. Furthermore, a unique feature of MYO18A is the presence of a PDZ domain at its N-terminal extension [152,153]. Neither MYO18A nor MYO18B can be considered true actin motors, since they lack ATPase activity and function primarily as scaffolding proteins [152]. Since MYO18A and MYO18B have long C terminal tails resembling those of conventional myosins, they co-assemble with NM II in cell-free systems [154] and regulate formation of high order contractile NM II structures, such as stress fibers in cultured cells [155].

Several studies highlight MYO18B as a tumor suppressor, which is mutated, deleted or methylated in colorectal, lung and ovarian cancer (Table 1) [60,156,157]. The notion of the tumor-suppressing activities of MYO18B is supported by studies involving overexpression of this cytoskeletal scaffold in colon and lung cancer cell lines that resulted in suppression of anchorage-independent cancer cell growth (Figure 2) [60,156]. However, mechanisms of growth-suppressing effects of MYO18B have not been elucidated.

The roles of another member of class XVIII myosins, MYO18A, in oncogenesis remain poorly characterized. A recent study identified frequent MYO18A mutations in stage III CRC [59]. These mutations appear to be an independent prognostic factor of clinical outcomes and were significantly associated with increased disease-free patient survival [59]. Since the CRC-associated mutations of MYO18A have not been linked to altered expression or activity of this protein, it is difficult to establish whether their beneficial effect is due to the loss- or gain-of-function of MYO18A.

A different group of evidence suggests that this myosin scaffold may contribute to tumor development and metastasis by regulating activity of its major binding partner, Golgi phosphoprotein 3 (GOLPH3). GOLPH3 is a phosphoinositide-binding protein that regulates Golgi morphology and vesicle trafficking [158]. At the Golgi, GOLPH3 binds to MYO18A and such binding is essential for proper GOLPH3 localization as well as Golgi integrity [159]. GOLPH3 appears to be an oncogene upregulated in many types of cancer, including CRC [158,160,161]. In CRC, high GOLPH3 expression significantly correlates with poor patient survival and high tumor recurrence [160,161]. Since MYO18A serves as a critical regulator of GOPLH3 localization and functions, it could be essential for the oncogenic activity of this protein. Interestingly, DNA-damaging anticancer drugs trigger GOLPH3 phosphorylation, which increases the strength of the GOLPH3–MYO18A interactions [162]. Furthermore, depletion of either GOLPH3 or MYO18A decreases cancer cell survival following DNA damage [162]. This data highlights the MYO18A–GOLPH3 complex as an important modulator of the efficiency of anticancer therapy.

## 4. Pharmacologic Modulators of Myosin Activity: Are They Suitable for the Development of Anticancer Drugs?

The well-recognized roles of various members of the myosin superfamily in driving cancer cell growth and motility raise the important question of whether small molecules that modulate activity of these cytoskeletal motors could be used as tumor-suppressing and anti-metastatic drugs. Several small molecules targeting different conventional and unconventional myosins have been developed and some of them have been extensively used as analytical tools to probe cellular functions of these motors in vitro [163,164,165]. By contrast, possible therapeutic implications of pharmacologic myosin modulators in vivo remain poorly explored. Possible reasons for the lack of investigation include a common perception of myosins as poorly “druggable” targets due to their important homeostatic functions in the heart, skeletal muscle and blood vessels, as well as the unfavorable physico-chemical characteristics of the most commonly used myosin inhibitors. Recent studies led to significant improvement and expansion of myosin-targeting pharmacologic tools. While very few studies to date examined the potential anticancer activities of these compounds in vivo (Table 2), a growing molecular diversity and better characterization of their targets increase perspective for future therapeutic applications of pharmacological myosin modulators.

### 4.1. Anticancer Effects of Small Molecular Modulators of Conventional Myosins

Small molecular inhibitors of conventional myosins have been known for more than three decades, although application of the initial inhibitors, such as 2,3 butanedione monoxime, were limited due to their low affinity and multiple off-target effects [164]. Discovery of blebbistatin, a cell-permeable compound with significant specificity toward class II myosins [176], provided a unique analytical tool to investigate myosin-dependent processes in live cells and organisms [165]. Blebbistatin blocks all conventional myosin motors by interfering with their mechanochemical cycle and locking them in the actin-detached stage [165]. This compound has been commonly used to probe NM II functions in regulating growth and motility of different mammalian cells, including cancer cells in vitro [41,45,106,111,114,177]. Furthermore, blebbistatin demonstrated anti-inflammatory [166], tissue protective [168,169] and anti-thrombotic [167] effects in animal models in vivo. Virtually nothing is known about the potential tumor suppressing effects of blebbistatin in mouse models of tumorigenesis, with only one study reporting reduced central nervous system infiltration of engrafted leukemia cells in blebbistatin-treated animals [170]. Two major problems hamper blebbistatin usage as a therapeutic anti-cancer agent. One is its broad selectivity to different conventional myosins, which may result in various cardiac, muscle and vascular side effects. The second problem is that this compound has unfavorable physico-chemical properties, including poor solubility in aqueous solutions, light sensitivity and phototoxicity [165,178]. Recently, more stable and less cytotoxic derivatives of blebbistatin have been synthesized [178,179,180]; however, the biological actions of these derivatives in vitro and in vivo remain poorly characterized.

Accumulating evidence about NM II inhibition in metastatic cancer cells [15,115,171,181] prompted a search for small molecular NM II activators with anti-metastatic properties. This search resulted in identification of 4-hydroxyacetophenone (4-HAP), a compound that activates NM IIB and NM IIC isoforms by increasing the assembly of their heavy chains independently of RMLC phosphorylation [171,181]. By contrast, 4-HAP does not alter the biochemical properties of the NM IIA paralog. Activation of NM IIB and NM IIC with 4-HAP increases stiffness and reduces adhesion and migration of pancreatic and colon cancer cells in vitro [115,171]. Importantly, this NM II activator attenuates metastasis of pancreatic and colon cancer cells engrafted in nude mice [115,171]. Despite these encouraging data, it remains unclear whether 4-HAP could serve as a promising pro-drug for CRC-targeting therapy. Indeed, this compound does not affect NM IIA, which is the most important and abundantly expressed NM II paralog implicated in CRC development. Conversely, little evidence exists to support roles of the proposed 4-HAP targets, NM IIB and NM IIC, in colorectal tumorigenesis. EMD 57033, a chemical chaperone with broad specificity to different conventional myosins, has been described [172,182], although its effects on cancer cell behavior have not been investigated. Cumulatively, the studies of the NM II-modulating compounds reviewed above demonstrate that either pharmacologic inhibition or activation of these conventional myosin motors could inhibit tumor growth and metastasis. This is consistent with the complex roles of NM II in regulating the tumorigenesis and highlights the critical need for better understanding the activation status of these actin motors at different stages of CRC development.

### 4.2. Small Molecular Inhibitors of Unconventional Myosins

In contrast to conventional myosins, small molecular modulators have not been extensively used to probe functions of unconventional myosins [163,164]. However, pharmacologic inhibitors have been discovered for three unconventional myosin classes I, V and VI that are known to be upregulated in CRC. For example, a marine antibiotic pentachloropseudilin (PCLP) was shown to inhibit motor properties of class I myosins in the low micromolar range with a much higher potency, as compared to inhibition of class II and V myosins [183]. Cellular activities of PCLP appear to be consistent with the known functional roles of class I myosins in regulating vesicle trafficking and actin cytoskeleton architecture. Thus, PCLP treatment perturbed lysosomal morphology in HeLa cells [183], as well as altered the endocytic trafficking and recycling of transforming growth factor (TGF)-β receptor and integrins in various human cancer and primary endothelial cells [184,185]. Furthermore, PCLP caused abnormal cytokinesis and cortical F-actin remodeling in zebrafish embryos [173] and inhibited the early morphogenic steps in chick embryos [174]. Perturbed protein trafficking and cytoskeletal organization appears to mediate PCLP-dependent inhibition of TGF-β-induced epithelial-to-mesenchymal transition and motility of cancer cells [184], as well as attenuated extracellular matrix adhesion and migration of endothelial cells [185]. These examples highlight the potential antimetastatic properties of myosin I inhibitors.

A search for small molecules that could block activity of class V myosins identified two different inhibitors, MyoVin-1 [186] and pentabromopseudilin (PBP) [187]. Both small molecules inhibit myosin V ATPase activity in a low micromolar range (IC50 ~ 1.2–6 μM) and have a much lower inhibitory activity toward conventional and some other unconventional myosin classes in a biochemical assay [186,187]. Limited data is available about the biological activity of these compounds in cells and tissues. Both MyoVin-1 and PBP impair vesicle trafficking in stimulated rat hippocampal synapses [188]. Furthermore, PBP blocked TGF-β-dependent cytoskeletal remodeling, mesenchymal transdifferentiation and migration of lung adenocarcinoma cells [189]. These inhibitory effects of PBP were associated with reduced cell surface expression and increased degradation of TGF-β receptor [189]. Since MyoVin-1 and PBP are likely to inhibit all members of the class V myosins and the reported opposite roles on MYO5A and MYO5B in CRC, the prospects for therapeutic use of class V myosin inhibitors in treating colon cancer remain unclear.

A series of computational and biochemical studies discovered a halogenated phenol, 2,4,6-triiodophenol (TIP) as selective inhibitor of the MYO6 motor [190]. TIP is a relatively potent MYO6 inhibitor with an IC50 of approximately 2 μM. Even at a much higher level (50 μM), TIP did not significantly affect the ATPase activity of NM IIC, cardiac myosin II and MYO1D [190]. Consistent with known MYO6 functions, TIP was shown to inhibit fusion of exocytic vesicles with plasma membrane in HeLa cells [190]. Importantly, TIP treatment blocks synaptic plasticity in cerebellar slices of control, but not in MYO6 knockout mice, a finding which confirms the target-specific effects of this inhibitor ex vivo [191]. However, before its identification as a MYO6 inhibitor, TIP (also known as Bobel-24 or AM-24) was found to block leukotriene B4 (LTB4) synthesis in human volunteers [175]. It was shown to induce caspase-independent death of pancreatic adenocarcinoma and leukemia cells in vitro [192,193], which at the time was explained as a perturbation of eicosanoid synthesis. LTB4 is known to play essential roles in tumor development [194]. Therefore, the dual inhibitory actions of TIP on MYO6 activity and on LTB4 production could interfere with two independent mechanisms of tumorigenesis, thereby providing an attractive possibility of using this compound for the development of novel anticancer treatment in CRC.

## 5. Conclusions

Altered organization of the actin cytoskeleton is a critical mechanism of CRC development that fuels the growth and metastatic dissemination of cancer cells. This complex process involves multiple players, among which myosin motors play key roles by controlling various stages of oncogenesis, from nuclear transcriptional programing to cell cortex remodeling in migrating and dividing cancer cells. Accumulating evidence highlights several members of the myosin superfamily as either tumor promoters or tumor suppressors in CRC. These contrary effects may reflect a perturbed balance of various actin motors, which control cellular homeostasis during oncogenesis. While altered expression of some conventional and unconventional myosins have been highlighted as possible biomarkers of disease progression and outcome, a precise picture of myosin actions and functions during CRC development is lacking. How various myosins control the interplay among the autonomous cytoskeletal remodeling of cancer cells, alter the physical properties of the tumor microenvironment and affect tissue inflammation, all remain poorly understood. Finally, developing small molecular modulators of myosin activity as therapeutic anticancer agents is an exciting but underdeveloped field of research.

## Figures and Tables

**Figure 1 cancers-13-00741-f001:**
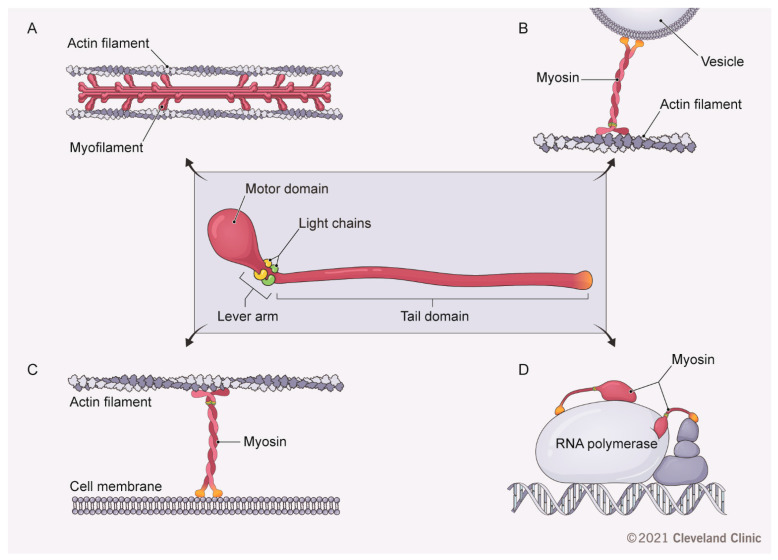
Molecular organization and cellular functions of myosin motors. The diagram illustrates a common domain structure of different conventional and unconventional myosins and their most characterized cellular activities in regulating actin filament sliding/contractility (**A**), vesicle trafficking along actin filaments (**B**), tethering actin filaments to cellular membranes (**C**) and assembly of nuclear transcriptional complexes (**D**). Figure 1 by Brandon Stelter, BFA. Reprinted with the permission of the Cleveland Clinic Center for Medical Art & Photography © 2021. All Rights Reserved.

**Figure 2 cancers-13-00741-f002:**
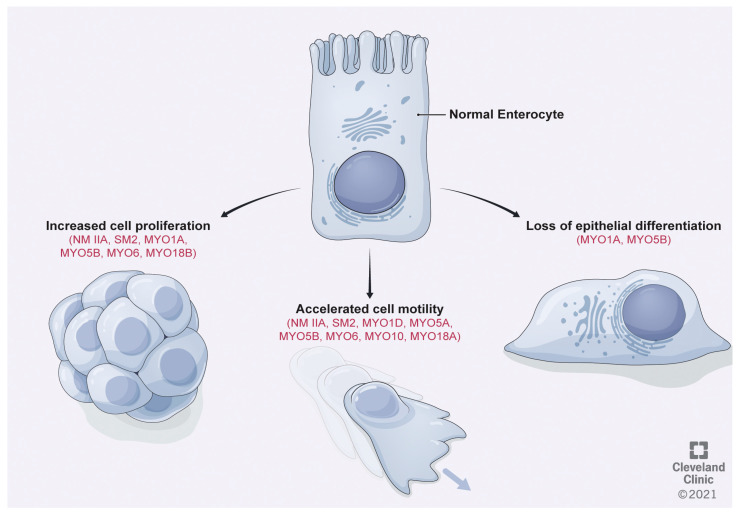
Functional effects of abnormal myosin activities in colon cancer cells. This figure depicts the known or proposed functional effects of altered expression or activity of different myosins in colon cancer cells that include increased cell proliferation, accelerated motility and loss of epithelial differentiation. See the text for abbreviations of different members of the myosin superfamily. Figure 2 by Brandon Stelter, BFA. Reprinted with the permission of the Cleveland Clinic Center for Medical Art & Photography © 2021. All Rights Reserved.

**Table 1 cancers-13-00741-t001:** Altered expression of different myosin motors in clinical colorectal cancer (CRC).

Myosin	Expressional Changes	mRNA or Protein Analysis	Prognostic Value	References
Non-muscle myosin IIA (*MYH9*)	Upregulated	mRNA and protein level	High protein expression is associated with poor overall survival	[43,44,45]
Non-muscle myosin IIA	Upregulated in CRC stroma	Protein level	No data	[46]
Smooth muscle myosin isoform 2 (*MYH11*)	Mutated and upregulated	Genomic sequencing; mRNA and protein expression	High protein expression is associated with poor overall survival	[47,48,49,50]
Myosin-1A (*MYO1A*)	Mutated and downregulated	Genomic sequencing; protein expression	Low protein expression is associated with poor overall survival and disease-free survival	[51]
Myosin-1D (*MYO1D*)	Upregulated	Protein level	No data	[52]
Myosin-5A (*MYO5A*)	Upregulated in metastatic CRC	mRNA expression	No data	[53]
Myosin-5B (*MYO5B*)	Downregulated	mRNA and protein levels	Low mRNA expression is associated with poor overall survival and disease-free survival	[54]
Myosin-6 (*MYO6*)	Upregulated	mRNA and protein levels	mRNA overexpression is associated with poor overall survival	[55,56,57]
Myosin-10 (*MYO10*)	Downregulated	mRNA expression	No data	[58]
Myosin-18A (*MYO18A*)	Mutated	Genomic sequencing	Mutations are associated with better disease-free survival	[59]
Myosin-18B (*MYO18B*)	Downregulated	mRNA expression	No data	[60]

Note: Names of the human myosin genes are presented in parentheses.

**Table 2 cancers-13-00741-t002:** Cellular targets and biological effects of the pharmacological modulators of conventional and unconventional myosins.

Compound	Protein Target	Mechanisms of Actions	Effects in Non-Cancer Models In Vivo	Effects in Cancer Models In Vivo
Blebbistatin	NM II and other class II myosins	Locks the myosin ATPase cycle in the actin detached state	Renal protective and anti-inflammatory effects in a rat model of nephropathy [166]. Inhibits development of carotid arterial thrombosis in mice [167]. Neuroprotective effects in the cerebral ischemia/reperfusion model in mice [168,169].	Inhibits central nervous system infiltration of leukemia cells in mice [170]
4-Hydroxyacetophenone	NM IIB and NM IIC	Increases myofilament assembly	No data	Inhibits metastasis of colon and pancreatic cancer xenografts [115,171]
EMD 57033	Class II myosins	Promotes myosin folding and ATPase activity	Improves functions of the canine failing heart [172].	No data
Pentachloropseudilin	Class I myosins	Inhibits myosin motor activity by uncoupling the actin and ATP binding sites.	Impairs cytokinesis and cortical actin filament assembly in Zebrafish embryos [173]. Impairs early morphogenesis in chick embryos [174]	No data
Pentabromopseudilin	Class V myosins and in less extent, class II myosins	Inhibits ATPase activity.	Impairs early morphogenesis in chick embryos [174].	No data
MyoVin-1	Class V myosins	Inhibits the ATPase cycle by blocking ADP release from myosin.	No data	No data
2,4,6-triiodophenol	Class VI myosins	Inhibits actin-stimulated ATPase activity.	Inhibits LTB4 synthesis in human healthy volunteers [175]	No data

## Data Availability

No new data were created or analyzed in this study. Data sharing is not applicable to this article.

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
