# Peer review of "Myosin Motors: Novel Regulators and Therapeutic Targets in Colorectal Cancer"

_cancers, 2021, doi:10.3390/cancers13040741_

Round 1

Reviewer 1 Report

Naydenov et al. reviewed the literature on the subject ‘Myosin motors: new regulators and therapeutic targets in colon cancer’. This is a very interesting and often overlooked topic, and the actin cytoskeleton and ABPs are an excellent therapeutic target in many cancers.

Therefore, I suggest to re-structure the manuscripts as follows:

  • General: Please explain all abbreviations   (i.e K-RAS, ACP, VASP, etc).
  • use the commonly used abbreviation ABPs (Actin Binding Proteins)
  • In the introduction, please describe in detail the actin cytoskeleton (3-4 additional sentences about the structure)
  • In chapter 2.1, the RMLC abbreviation is explained as regulators myosin light chains, while e.g. in line 185 the abbreviation RLMC is used which may mislead the reader, and as I understand it is the same
  • Mechanisms of CM inactivation in colon cancer. What does the abbreviation CM mean, as there is no explanation?
  • In the table please explain ND abbreviation
  • Finally, please explain why is it difficult to manipulate actin and ABPs? Due to the crucial role of actin and myosin in the heart and skeletal muscles.

The article is interesting, although a bit encyclopedic, and difficult because it contains a lot of information. In my opinion, however, it is valuable and I recommend the article to be accepted for publication after minor revisions.

Author Response

We thank the reviewer for a stating that the manuscript targets ‘a very interesting and often overlooked topic’ and is ‘valuable’.

Minor Comments

Comment 1: Please explain all abbreviations.

Response: We have either explained all abbreviations, or removed unnecessary abbreviations.

Comment 2: Use commonly used abbreviation ABPs (actin binding proteins).

Response: We appreciate this comment but believe that ABP is usually reserved to the proteins that regulate actin filament turnover (polymerization and depolymerization) and is not frequently used to describe myosin motors (although they are actin-binding proteins). Since we rarely mention actin filament turnover in this review paper and already use a large number of different abbreviation, we would prefer not to introduce another one.  

Comment 3: In the introduction, please describe details of the actin cytoskeleton.

Response: As recommended by the reviewer we added a couple of sentences to highlight organization and dynamics of the actin cytoskeleton.

Comment 4: There is a confusing usage of RMLC abbreviation.

Response: We apologize for this error that has been corrected.

Comment 5: Need to explain CM abbreviation.

Response: CM means ‘conventional myosins’. This abbreviation has been removed.

Comment 6: Need to explain ND abbreviation in the Table 1.

Response: This has been de-abbreviated.

Comment 7: Please explain why it is difficult to manipulate actin and ABP (myosins) from the therapeutic perspective.

Response: As recommended by the reviewer, we explain that this is due to important functions of the actomyosin cytoskeleton in the heart and skeletal muscles.

Reviewer 2 Report

This is a very comprehensive and up-to-date review on the roles of myosins in colorectal cancer development. Although there are numerous reviews touching aspects of myosin and disease, I’m not aware of any tackling myosin and cancer globally, let alone restricted to CRC. This review is timely and should be of interest to those wishing to gain an overview on the stand of the research on this topic. The authors have made a commendable effort to collect and organise a considerable amount of information. While there is nothing to criticize about the scope and the contents, I would recommend the authors to make an effort to turn the text easier to navigate:

-Consider economising on background information throughout the text. Some level of detail can be cut without compromising the main messages. To give an example: class VI myosins in section 3.3. It would suffice to mention that this myosin, unlike others, moves cargo towards the minus end of the actin filament. Those familiar with myosins will know, others will easily find that information if they need.

-The authors should screen the text to make it less “wordy” and redundant. For example “Myosin V is an ancient subfamily of unconventional myosins that dimerize to form two-headed motors..” could go as “Myosin V forms two-headed motors…” 

-Each section is a single block of text. Splitting the blocks into paragraphs will ease reading. For example, the introduction could be split at lines 56, 69 and 100 (but join 105 and 112).

-The various mechanisms converging in the regulation of conventional myosins (phosphorylation of RLCs, K-Ras, S100) and how they impact on CRC could be summarised in a figure. The authors shouldn’t be afraid of speculating by incorporating knowledge from non-human models and other tumors that may help explain the role of conventional myosins in CRC pathogenesis.

-The section on pharmacologic modulators is no doubt important. The information would be conveyed better if some technical details were presented in a table: compound, specificity, pharmacologic properties, biological properties, etc, away from the main text. This way the text would be lighter and better focussed on potential applications and drawbacks.

All together aim at making the paper 10-15% shorter without compromising the core information.

Minor

-Consider changing the title to “…targets in colorectal cancer”. The authors use CRC most of the time throughout the review.

-Line 71: give the exact number of myosin genes in human rather than an approximation.

-Line 88: it should be lever arm, not level arm. This error appears a few more times, please correct.

-Human gene names should go in upper case italics by convention. The authors are giving the mouse versions in the text when referring to the human genes. Please correct.

-Table 1.  Consider including the gene name in addition to the protein name.

-Line 153: an A769T point mutation.

-Line 185 should be RMLC, not RLMC. The same mistake occurs many times afterwards, please correct.

-Line 187: the zebrafish gene is myh11, lower case italics.

-Line 202: spell out AOM-DSS.

-Line 220: conventional myosin, not CM (you don’t use this abbreviation anywhere and it’s not needed)

-Line 228: mitogen-activated

-Line 435: pheochromocytoma

Author Response

Reviewer 2.

We thank the reviewer for characterizing our paper as ‘This is a very comprehensive and up-to-date review..’ and providing insightful comments that we address below:

Comment 1: Consider economizing on the background information by cutting on some levels of details.

Response:  We appreciate this comment and tried to remove unnecessary details from the manuscript

Comment 2: The authors should screen the text to make it less wordy.

Response: We followed this reviewer’s recommendation, to our best ability.

Comment 3: Splitting the text block into paragraphs is recommended

Response: Done, as suggested.

Comment 4: The various mechanism converging into regulation of myosin activity could be summarized in a Figure.

Response: We used a professional illustration service at Cleveland Clinic to prepare the final version of the Figures. Due to their very busy schedule it could take several months to prepare a new Figure, which is unfeasible for the required short turnaround of the manuscript resubmission.

Comment 5: The authors should not be afraid of speculating by incorporating knowledge from non-human models and other tumors that may explain the roles of conventional myosins in CRC.

Response: We appreciate this comments and believe that we already incorporated relevant knowledge from other tumors, non-tumor animal model and cell culture experiments. We added a bit more data on this subject during the revision. Myosins are very extensively studied cytoskeletal proteins with huge amount of information amassed over last decades. We afraid that discussing too much data will complicate navigating through this review, which is already views as ‘a bit encyclopedic and difficult’ by Reviewer 1 and recommended to be shortened by Reviewer 2.

 Comment 5: Some properties of pharmacological modulators of myosin activity could be summarize in a table.

Response: We appreciate this suggestion and now summarize the data regarding pharmacological modulators of myosin activity in a new Table 2.

Comment 6: All together aim at making the paper 10-15% shorter.

Response: As recommended, we tried to implement more economic writing to shorten the manuscript.

Minor comments:

  • Consider changing the title to ..targets in colorectal cancer’

Response:  We have modified the title, as recommended

  • Line 71: give the exact number of human myosin genes.

Response:  Done as suggested.

  • Line 88: It should be lever arm, not level arm.

Response:  Apology for this error, which has been corrected.

  • Human gene name should be in upper case italics convention. The authors give the mouse version in the text when referring to human genes.

Response:  This has been corrected.

  • Table 1. Consider including the gene name in addition to the protein name.

Response:  The gene name has been included in the Table 1.

  • Line 153: an A769T mutation.

Response:  Corrected, as requested.

  • Line 185: should be RMLC, not RLMC.

Response:  Corrected, as requested.

  • Line 153: the zebrafish gene is myh11, lower case

Response:  Corrected, as requested.

  • Line 202: spell out AOM-DSS.

Response:  Done.

  • Line 220: De-abbreviate CM.

Response:  Done.

  • Line 228: mitogen-activated.

Response:  corrected

  • Line 435: pheochromocytome.

Response:  Corrected

Reviewer 3 Report

The article review "Myosin Motors: New Regulators and Therapeutic Targets in Colon Cancer" by Naydenov et al.  is update and well written. The topic is current as the knowledge of the role of altered cytoskeleton and, consequently, of the myosin motors in carcinogenesis is constantly evolving. The “introduction” section on the molecular organization of the myosin superfamily members is clear and exhaustive. Equally detailed is the section focusing on the activation status and functional roles of conventional and unconventional myosins in colon cancer. However, the article appears unbalanced because the final section on the development of pharmacological tools is not as detailed as the previous ones. The article is a bit lacking, in my opinion, in this section which is the real novelty of the review, namely the potential therapeutic applications in oncology related to the members of this protein superfamily. This so crucial aspect, also mentioned in the title, should be addressed in more detail as it is of great clinical interest.

Author Response

Reviewer 3.

We thank the reviewer for stating that the manuscript is ‘update and well written’ and providing an insightful comment that we address below:

Comment: The reviewer suggests that the article appears unbalanced since the pharmacological tools section is not as developed as other chapters. The reviewer suggest addressing this topic in more details

Response:  We appreciate this comment but would like to emphasize that this seemingly unbalanced composition of the review simply reflects the fact that the area of pharmacological manipulation of different myosins remain markedly underdeveloped, as comparing to significant knowledge accumulated by using genetic and imaging tools to visualize  and interrogate myosins functions in vitro and in vivo. Most importantly, almost nothing has been reported about modulating activity of different myosins by pharmacological tools in animal models of cancer. Even the best characterized inhibitor of conventional myosins, blebbistatin, was used at very few occasions as potential anti-cancer agent in vivo. Therefore, it is really difficult to expand this clinically important subject based on the available data. We now emphasize this paucity of cancer related data for pharmacological myosin modulators in the revised manuscript and summarize the pharmacologic myosin modulator data in a new Table 2.

Please also note that both reviewers 1 and 2 consider this manuscript as somewhat too long and encyclopedic, which argues against further expansion of certain discussion points of the paper.

Associate Editor’s Comments

Comment: This a very comprehensive and up-to-date review on the roles of myosins
in colorectal cancer.
The reviewers have suggested modifications in the presentation and
organization of the review.
The final section on the development of pharmacological tools should be
also more details as it is an important part of the scope of the review.
We encourage also the authors to make their own original figures, as the
only 2 figures of the review are reprints from other source."

Response: We appreciate these comments. The manuscript have been revised according to the reviewers’ recommendations.

The issue with the pharmacologic tools section has been addressed in our responses to Reviewer 3 comments.

The manuscript includes two original Figures designed by the authors and prepared by professional service at the Cleveland Clinic Center for Medical Art & Photography. According to the Cleveland Clinic rules, these Figures are copyrighted, but we obtained permission to reproduce them in our paper.

Round 2

Reviewer 2 Report

The authors have addressed my critique adequately. It is a pity a summary figure as suggested (comment 4) cannot be prepared for understandable reasons. Comment 5 referred actually to incorporating knowledge from other tumors etc to the suggested figure. I'm aware of those details already being discussed in the text. Apologies if I wasn't clear.